# Cyanobacterial Blooms and the Presence of Cyanotoxins in the Brazilian Amazon

**DOI:** 10.3390/toxins17060296

**Published:** 2025-06-11

**Authors:** Maria Paula Cruz Schneider, Elane Cunha, Lucas Silva, James Leão, Vanessa Costa Tavares, Eliane Brabo de Sousa, Silvia Faustino

**Affiliations:** 1Laboratory of Genomics and Biotechnology, Biological Sciences Institute, Federal University of Pará, Belém 66075-110, PA, Brazil; silvalucas.ldss@gmail.com; 2Laboratory of Taxonomy and Ecology of Cyanobacteria and Microalgae, Institute of Scientific and Technological Research of the State of Amapá, IEPA, Macapá 68903-419, AP, Brazil; elanedsc@yahoo.com.br; 3Federal Institute of Pará, Belém 66050-160, PA, Brazil; james.araujo@ifpa.edu.br; 4Laboratory of Cyanobacteria and Aquatic Bioindicators, Environment Section, Evandro Chagas Institute, Ananindeua 66023-971, PA, Brazil; vanessacosta@iec.gov.br (V.C.T.); elianesousa@iec.gov.br (E.B.d.S.); 5Algae Cultivation Laboratory, Federal University of Amapá, Macapá 68903-419, AP, Brazil; fitomathes@yahoo.com

**Keywords:** Amazonian rivers, bloom-forming cyanobacteria, microcystin, toxins

## Abstract

The records of blooms and cyanotoxins in the Brazilian Legal Amazon are scarce and do not represent the reality observed in recent years when there was an increase in notifications and studies carried out in this region. In this article, we carefully analyzed 48 studies to identify the locations where the blooms occurred, the most frequent species, and the tests performed to detect cyanotoxins within the territory of the Brazilian Legal Amazon. The analyzed studies identified approximately 145 taxa of cyanobacteria, and the most frequent species belong to the genera *Microcystis* and *Planktothrix*. The most frequently detected cyanotoxin was microcystin, and, in some locations, even with a low cell density, there was the production of toxins. In most cases, *Microcystis* and *Dolichospermum* were the major genera associated with blooms and toxin production. The state of Pará had the highest number of records of toxin-producing cyanobacteria, including records of seasonal events, while Acre had no records. This work contributes to our knowledge of the geographic distribution and occurrence of cyanobacteria and cyanotoxins in the Brazilian Amazon and proposes new strategies for improving the monitoring of cyanotoxins in the Amazon region.

## 1. Introduction

The Brazilian Amazon is renowned for its biodiversity and is one of the most well-preserved regions on the planet. This biome spans approximately 5.2 million km^2^ and is traversed by more than 25,000 km of navigable rivers connecting diverse ecosystems, including igapós, dry land, and floodplains [1]. This natural complex accounts for around 61% of Brazil’s territory [2]. The Amazon Hydrographic Region, which makes up 40% of the country’s territorial expanse, has about 81% of its area covered by native vegetation, highlighting its global environmental significance [3].

However, the region faces increasing pressures from emerging development models and predatory, illegal economic activities, such as illegal mining [4], compounded by improper land use and occupation. Above all, the low coverage of basic sanitation in the Brazilian Legal Amazon municipalities [5] creates an environment vulnerable to cyanobacteria blooms, a phenomenon typical of eutrophic environments. Eutrophication diminishes water quality, and its use for recreation, fishing, and human consumption may be restricted, mainly when harmful cyanobacteria blooms occur, releasing toxins into the water [3].

Although cyanobacterial blooms are often associated with Brazil’s most urbanized, industrialized, and agricultural regions, such as the south and southeast regions, where the diffuse input of nitrogen and phosphorus leads to eutrophication, this phenomenon also occurs in the Amazon. However, these events are poorly studied and rarely reported in the Amazon region, primarily documented in reservoirs and less frequently in rivers.

The Amazon has a rich diversity of cyanobacteria thanks to its favorable climatic conditions, including high temperatures, luminosity, and humidity. Studies have identified several strains of cyanobacteria in Amazonian rivers, such as *Pseudanabaena*, *Leptolyngbya*, *Planktothrix*, and *Phormidium*, which are genetically distinct from other strains found in different regions of Brazil and the world [6]. We employed ecological models to predict areas susceptible to predict areas susceptible to cyanobacterial blooms in Brazil, but there are significant sampling gaps in the Amazon region [7]. A major factor that can increase the occurrence of blooms in the Amazon is deforestation, which leaves the soil unprotected and amplifies the effect of leaching as a nutrient source for aquatic environments [8].

The Legal Amazon, comprising several Brazilian states—Acre, Amapá, Amazonas, Maranhão, Mato Grosso, Pará, Rondônia, Roraima, and Tocantins (Appendix A)—has a significant information gap regarding the occurrence of cyanobacterial blooms. Few records are available and are mainly in scientific publications with regional circulation and in local repositories of universities and research centers. The latest bulletin from the Brazilian Ministry of Health indicated that this region is the least documented concerning blooms in public water supplies [9]. To illustrate the Amazonia gap of cyanobacteria and cyanotoxins information, the public online database “SISAGUA”, which provides information on Drinking Water Quality Surveillance, implemented by the Ministry of Health, with data access about water quality parameters, such as cyanobacteria and cyanotoxins in drinking water sources [10], shows a total of 36,892 registries for cyanobacteria or cyanotoxins for all Brazil, and only five registries are for northern Brazil (Amazon region); none of these give cyanotoxins information.

In 2019, an article on the history and geographic distribution of cyanotoxins and cyanotoxin poisoning worldwide identified 105 cyanotoxins in South America. However, no records exist for the Brazilian Amazon region (Figure 1) [11].

In this context, this study aimed to verify, through a literature review, the distribution of cyanobacterial blooms and toxins in the states that comprise the Brazilian Legal Amazon and to identify spatial–temporal patterns associated with the rise in cyanobacteria density. It also provides recommendations for cyanotoxin monitoring in the Brazilian Amazon.

## 2. Results

### 2.1. Literature Review

We reviewed 48 studies examining cyanobacteria and toxins in the Amazon, comprising 24 research papers, six PhD theses, 13 master’s dissertations, two books, and three technical reports [12,13,14,15,16,17,18,19,20,21,22,23,24,25,26,27,28,29,30,31,32,33,34,35,36,37,38,39,40,41,42,43,44,45,46,47,48,49,50,51,52,53,54,55,56,57,58,59] (see Figure 2 and Appendix A. In total, 145 species of cyanobacteria from 12 orders were identified (refer to Appendix A).

Of the nine states included in the Legal Amazon (Appendix A), only Acre has not reported any occurrences of blooms or cyanotoxins to date. Reports of blooms and cyanotoxins are displayed in Figure 3.

### 2.2. Blooms

The blooms in the region primarily consisted of species from the genera listed in Table 1. The genera *Microcystis* and *Planktothrix* were the most common, identified in seven and six regional states, respectively. Figure 4 illustrates some of these species, and a record of two blooms [46,60] in the state of Pará is shown in Figure 5.

Remote sensing analyses performed using the CyanoLakes platform allowed the spatial–temporal identification of historical bloom events on the Tapajós River in Santarém, Pará, between November 2018 and October 2019 (Figure 6a,b). The phenomenon occurred mainly at the beginning of the rainy season and, curiously, between two regional conservation units. The bloom fluctuation corridor followed the river’s flow on the left bank mainly during the river flood, with a considerable reduction in the dry period (Figure 6c). During the period when rainfall began to increase, and the river’s water levels rose, the bloom concentrated at the mouth near the Amazon River.

### 2.3. Cyanotoxins

Cyanotoxins were registered in five states: Amapá, Pará, Roraima, Rondônia, and Tocantins. The methods employed included HPLC, ELISA, and toxicological assays involving rats. Of the 48 studies analyzed in this review, only 12 detected toxins (Table 2). Ten reported also cyanobacteria blooms in the UHE reservoir in Tocantins [59], in the Iriri River [47], Bolonha Lake [23,38], Tapajós River [24,29,42,44,48], Igarapé Grande [54], and in the fishing environment of Amapá [13]. The two remaining studies indicated the presence of cyanotoxins in low densities of cyanobacteria [14,15,50], measured by cell count, biovolume, or chlorophyll-a concentration.

## 3. Discussion

The most common bloom-forming genera for the Brazilian Legal Amazon were *Microcystis* and *Planktothrix*; they are also very common in Latin American countries [61]. Both produce microcystin, the most common cyanotoxin found in the papers (Table 2).

The first record of a cyanobacterial bloom in the Brazilian Amazon occurred in the Tapajós River, located in Pará state, in November 1947 [36]. The dominant cyanobacteria belonged to the genus *Dolichospermum* sp. (formerly *Anabaena* sp.), with densities reaching up to 4650 individuals per liter. It primarily developed in the upper water layers, peaking at 5000 organisms per liter at a depth of 2.5 m.

The blooms of *Dolichospermum* sp. are historical and occur regularly, seasonally, in the Tapajós River, located in the western part of the state of Pará, near the city of Santarém. Bloom events mainly happen at the end of the dry season or the beginning of the rainy season [22,24,26,29,30,36,37,42,44,48]. The remains of the blooms sink and accumulate as layers of highly soft mud, three meters or more thick [37]. There are also historical records from 1948 of *Dolichospermum* sp. blooms in lakes near the Tapajós River, such as Lake Muretá and Caxambú; the latter was described as the “most cyanophytic of all,” with a cyanobacteria density of 7500 org∙mL^−1^ [36].

Until 2023 [62], there was no scientific consensus about the quantitative definition of the term “cyanobacteria bloom,” so the Amazonian bloom papers mentioned the visualization of blooms, the green mats, with a previously considered low cyanobacteria density, such as 3255 cells.mL^−1^ [22], if considering the thresholds of the Brazilian Ministry of Health [63] or WHO Alert levels [64]. However, after the proposal of a numerical definition of bloom for different types of water [62], we have numerically confirmed what the scientists described since the 1940s as blooms in Amazonian water bodies with cell densities above 1615 cells.mL^−1^, chlorophyll-a > 0.32 µg∙mL^−1^ or biovolume > 0.57 mm^3^∙mL^−1^ [62]. Another recent study reviewed Latin American papers on cyanobacterial blooms and concluded that 47% of the papers classified cyanobacterial blooms as having cell densities above 2000 cells.mL^−1^ [61], corroborating the criteria adopted in the present study.

Regardless of the blooms, we also found records of cyanotoxins in low cell densities: 2.1 μg∙mL^−1^ microcystin-LR at the mouth of the Amazon River with 1090 cells.mL^−1^ of cyanobacteria [14,15] and 0.75 µg∙mL^−1^ microcystin in Rondônia with 1030 cells.mL^−1^ [50].

The low number of toxin records shown in Table 2 generally does not result from negative outcomes in toxin tests. Instead, they reflect a lack of equipment and kits and the disparity between the vast territorial area and the limited number of specialists available. This situation also occurs in all Latin America [61].

The Brazilian Amazon boasts rich biodiversity, yet scientific research and monitoring resources remain scarce. Numerous articles have estimated the region’s phytoplankton biomass using remote sensing to gauge chlorophyll-a, but few have also incorporated data on biovolume or cyanobacteria counts. For this reason, we recommend that, as a strategy for the Amazon region, monitoring at drinking water intake points be done through cyanotoxin analysis rather than through the identification and counting of cyanobacteria or chlorophyll a. Cyanotoxin analyses can be performed by ELISA, which requires the purchase of an ELISA reader and washer, and then ELISA kits. These are rapid and straightforward methods that allow the processing of a large number of samples in a short time. Aguillera et al. [61] also recommend prioritizing toxin analyses by the ELISA method. The Ministry of Health’s ordinance [63] also permits the direct analysis of toxins, eliminating the need to identify and count cyanobacteria and chlorophyll. We believe that this is the best strategy for the Amazon region, corroborated by [61]. This recommendation will overcome the need for specialized technicians to identify genera and count cyanobacteria cells.

The rise in cyanobacterial blooms in the Tapajós River and the Pará River estuary in the Amazon region of Pará results from complex and multifactorial phenomena. Despite having distinct hydrological characteristics, the river basins of these environments share notable similarities in land use and occupation, marked by intense anthropogenic impacts over recent decades. These factors, possibly combined with climate change, have significantly contributed to the increase in these blooms.

The Tapajós River originates at the confluence of the Juruena and Teles Pires rivers in Mato Grosso state and flows into the Amazon River [65]. Cyanobacterial blooms occur seasonally, as noted, and frequently on the left bank, across from the municipality of Santarém [26,29]. In Santarém, within Lago Grande do Curuai, a vast floodplain region, blooms have also been documented [28,66]. The predominant genera in these areas are *Microcystis*, *Dolichopermum*, and *Planktothrix*, which are known to produce microcystin-LR [24,29].

The hydrogeochemistry of the Tapajós River is typical of clear waters [67], characterized by high transparency (>2 m), slightly alkaline pH, and low levels of turbidity, electrical conductivity, and dissolved solids concentration [68]. However, the river exhibits high concentrations of iron and aluminum due to the geological composition of the area’s rocks and soils [66]. Seasonal variations in water levels—flood (from December to March), high water (from April to June), low water (from July to September), and dry water (from October to November)—directly affect the composition and density of phytoplankton, including cyanobacteria, as well as water quality [28,69].

Blooms in the Tapajós River are more frequent during flooding and high-water periods [24] than in other periods of the year. In contrast, in Lago Grande do Curuai, the largest cyanobacterial biomass occurs during low-water and dry periods, with higher temperature, water stability, nutrient concentrations, and a more alkaline pH [27,28].

Studies have indicated an increase in nitrogen compounds and total particulate phosphorus in areas close to the most urbanized regions of the Tapajós River, mainly due to inadequate basic sanitation and sanitary infrastructure in the municipality of Santarém. This municipality is among the worst in Brazil concerning sanitation, notably regarding domestic sewage treatment [5]. Furthermore, climate change and alterations in the seasonal hydrological regime have intensified blooms. The recent example (see Figure 6f) occurred in early 2025 when, following an intense drought, a proliferation of cyanobacteria was recorded in Alter do Chão, a tourist village in Santarém [48].

The Pará River estuary is a body of water that originates in the Baía das Bocas, where part of the Amazon River is diverted to the Pará River through the Breves Strait, a series of long and deep rivers. It flows south of Marajó Island, receiving contributions from the Tocantins River, its primary tributary, and other smaller rivers [70]. Unlike the Tapajós River, the hydrogeochemistry of the Pará River estuary resembles that of whitewater rivers [67], exhibiting low transparency, highly variable pH (slightly acidic to alkaline), high turbidity, elevated electrical conductivity, and significant concentrations of total dissolved solids [71,72]. These waters contain high levels of iron and aluminum [73], similar to the Tapajós River.

During dry periods, the Atlantic Ocean exerts a more significant influence on the estuary, increasing salinity and altering the chemical composition of the water. Semidiurnal mesotidal tides dominate this estuary, reaching a maximum height of 4.2 m depending on the season. This dynamic creates strong currents and displaces large volumes of water [73], significantly affecting the local ecology and sedimentary processes. The combination of strong currents, high concentrations of suspended sediments, and the influence of seasonal climate makes the Pará River estuary one of the most dynamic hydrological systems in the Amazon region of Pará [71,73], standing in stark contrast to the calm and clear waters of the Tapajós River.

However, in this dynamic environment, cyanobacterial blooms also occur (Figure 5a, aerial photograph of the bloom), primarily involving species from the genus Microcystis: *Microcystis aeruginosa*, *Microcystis protocystis*, and *Microcystis wesenbergii* (Figure 4a–d). Unlike the Tapajós River, which has a record of blooms dating back to the 1940s, the blooms in the Pará River seem to be a more recent phenomenon that has been intensifying each year. Despite there being no consensus on their causes, it is of general knowledge that the city of Barcarena at the banks of the Pará River has intense industrial and port activities that affect the water quality of the rivers in this region.

The Tapajós River and the Pará River estuary share several significant anthropogenic impacts, including the lack of basic sanitation in the municipalities along their banks, deforestation, mining, intensive agriculture, industrial activities, a busy port region, and illegal mining. In the case of the Pará River, this illegal mining originates from the Tocantins River, its main tributary [3].

Environmental accidents are notably frequent in the estuary of the Pará River, primarily linked to port activities and the industrial complex in Barcarena. This industrial complex hosts companies that process and export kaolin, alumina, aluminum, and cables, in addition to firms that produce cellulose, fertilizers, and more [74]. These factors render the region especially susceptible to environmental impacts, including the potential proliferation of cyanobacteria.

Cyanobacterial blooms in the region have attracted attention from the local population and the media, establishing them as a widely recognized issue. Simultaneously, public health institutions in the state have played a crucial role in examining the effects of these blooms on water quality and health risks.

## 4. Conclusions

After revising 48 papers on cyanobacteria and cyanotoxins over the Brazilian Amazon, we conclude that there are three patterns for the rise of cyanobacteria densities in the Amazon Region. There is one pattern for rivers and reservoirs, another specifically for the Tapajós River near Santarém, and another for the Amazonian floodplain lakes. The cyanobacteria tend to rise in number during the transition period from rainy to dry season (from June to August–September) in rivers and reservoirs. The Tapajós River near Santarém has a pattern of cyanobacteria rising that matches the rainy season (from the beginning of December until April). Finally, floodplain lakes like Lago Grande de Curuai need more research to reach conclusions, and we recommend a specific review paper.

This study aimed to fill a knowledge gap concerning cyanobacterial blooms and cyanotoxins in the Amazon. Although cyanobacterial blooms have been reported in the region since at least 1947, comprehensive studies on these bloom events and their toxin production have been lacking. Our review emphasizes the influence of Amazonian seasonality on the occurrence of blooms and the necessity of comprehending blooms’ spatial and temporal patterns. This information may be used by drinking water treatment plants, reservoirs, and others to diminish health impacts.

Another pressing concern is the potential impact of climate change on cyanobacterial blooms in Amazonian rivers. Rising temperatures, altered precipitation patterns, and increased frequency of extreme weather events—such as prolonged droughts—may create more favorable conditions for bloom proliferation. For example, the bloom recorded in Alter do Chão in early 2025, following an intense drought, underscores the link between climatic anomalies and cyanobacterial expansion. This fact reinforces the need for long-term studies to assess how climate change might exacerbate bloom frequency and intensity across the Amazon basin.

Given these findings, it is imperative to intensify efforts to systematically monitor cyanobacterial blooms and cyanotoxins across the Amazon, especially in areas distant from state capitals, where scientific investigations and environmental management policies are often scarce. Collaboration between environmental agencies, academic institutions, and local communities is crucial to developing strategies for mitigating the negative impacts of these blooms. Strengthening research, monitoring, and mitigation measures will be essential to address the growing challenges posed by cyanobacterial blooms in the region, particularly in the face of ongoing environmental changes.

## 5. Materials and Methods

For this work, a bibliographic review was conducted, adhering to the criteria of a systematic review, which involves a predefined set of criteria concerning the type of study [75]. We searched various databases, including Google Scholar [76], Scielo [77], and Periódicos Capes [78], as well as repositories of scientific institutions in the Brazilian Amazon [79,80,81,82,83,84,85,86,87].

The search combinations were: “Cyanophyta and Bloom and Amazon”; “Cyanobacteria and Bloom”, “Phytoplankton”, “Floração + cianobactérias”, “toxic cyanobacteria”, “cyanotoxins”, “cianotoxinas”, “cianobactéria tóxica”, and “cyanoHAB”.

Since the scientific data on cyanobacteria and cyanotoxins in the Amazon Region are scarce and dispersed, we also read papers to look for, in their references, more scientific studies on cyanobacteria blooms and cyanotoxins in the Region.

The selection process involved three stages: analyzing titles, reading abstracts, and conducting complete readings. The criteria for literature inclusion were published articles, theses, dissertations, books, conference presentations, and technical reports with information on cyanobacteria blooms, toxins, or both in water bodies in the Brazilian Amazon. Each article was analyzed, with those that did not report cyanobacteria blooms or toxins in their results being discarded.

As there is no scientific consensus on defining a cyanobacterial bloom, we adopted the quantitative bloom definition proposed by Frau [62] in revising papers on cyanobacterial blooms worldwide. In this context, we considered Amazonian water bodies as oligotrophic inland waters, and the thresholds for considering a cyanobacteria bloom in this study were cyanobacteria cell density > 1615 cells.mL^−1^, chlorophyll-a > 0.32 µg∙mL^−1^, biovolume > 0.57 mm^3^∙mL^−1^ [62]. We also considered the papers containing qualitative detection, such as visual identification of greenish spots and/or the presence of foam. A recent review of papers on cyanobacteria blooms conducted in Latin America showed definitions of bloom varying from 2000 to 50,000 cells.mL^−1^, with 2000 cells.mL^−1^ being the most frequent definition of bloom, as reported in 47% of the papers [61]. Both results present similar quantitative numbers to define cyanobacteria blooms.

The spatiotemporal monitoring of the blooms for the Tapajós River was carried out using the CyanoLakes platform [88].

## Figures and Tables

**Figure 1 toxins-17-00296-f001:**
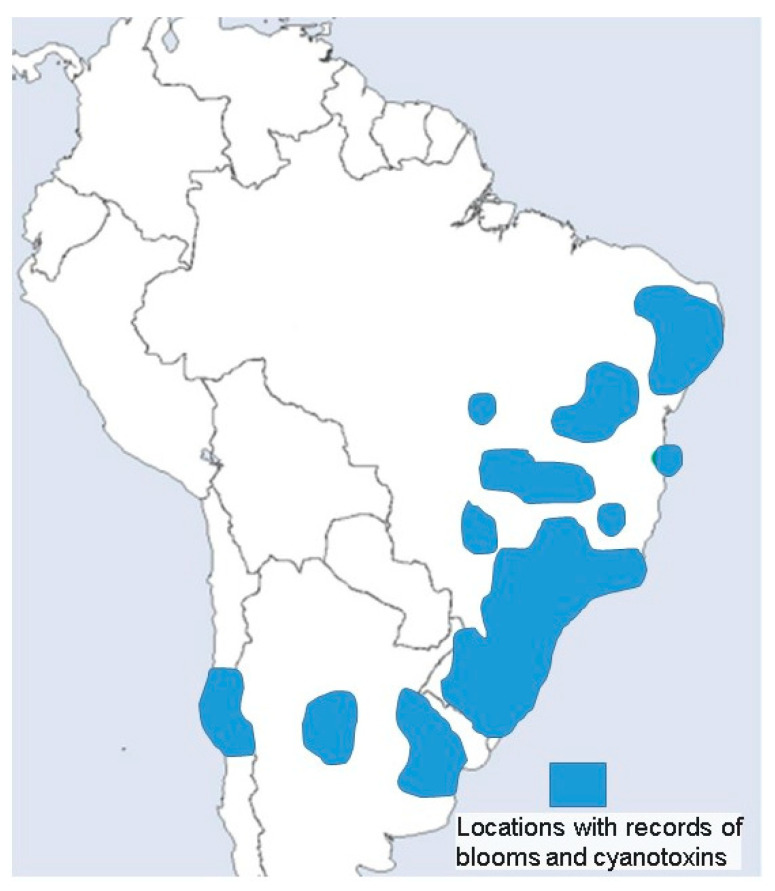
A map adapted from Svircev et al. (2019) [11] shows locations in South America with cyanobacterial blooms and toxins records. There are no records from the Amazon region.

**Figure 2 toxins-17-00296-f002:**
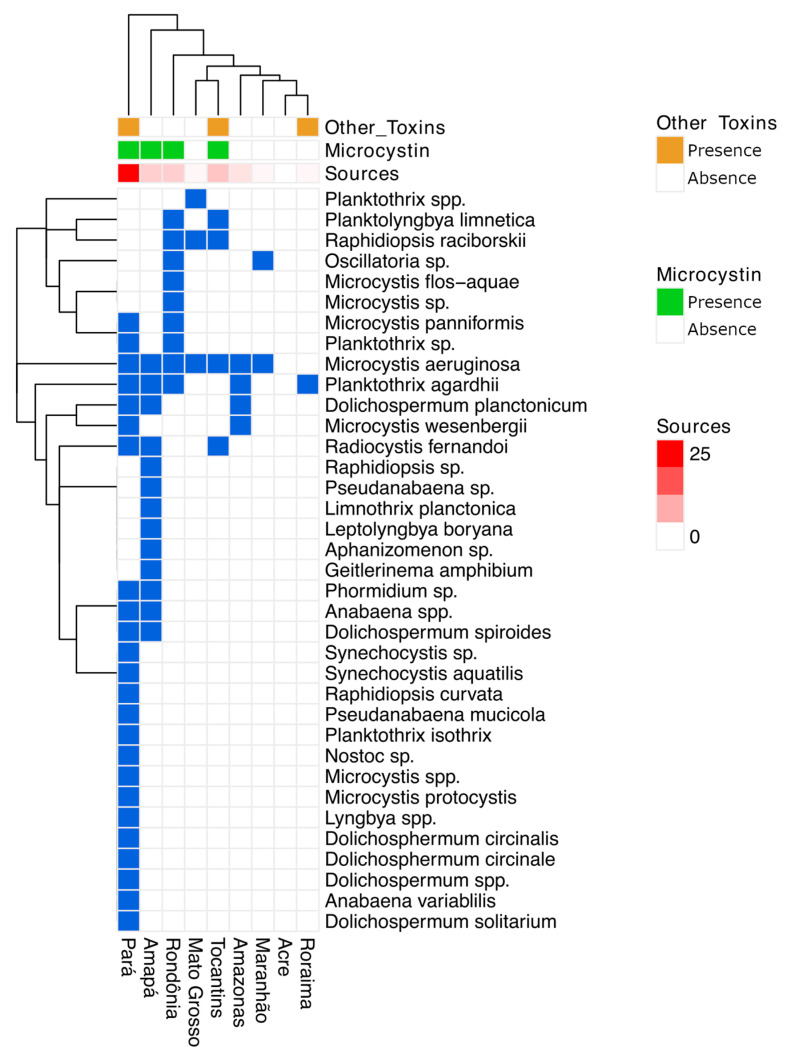
Heatmap showing the distribution of bloom-associated cyanobacteria in different states of the Brazilian Legal Amazon. Columns represent the states, while rows correspond to the species found. Blue-filled cells indicate presence, while empty cells indicate absence. The dendrogram on the left and at the top represents the hierarchical clustering analysis based on the similarity, grouping similar occurrence patterns. The top rows indicate the presence of microcystin (green) and other toxins (brown), as well as the number of sources used to obtain information, ranging from 0 (white) to 25 (red).

**Figure 3 toxins-17-00296-f003:**
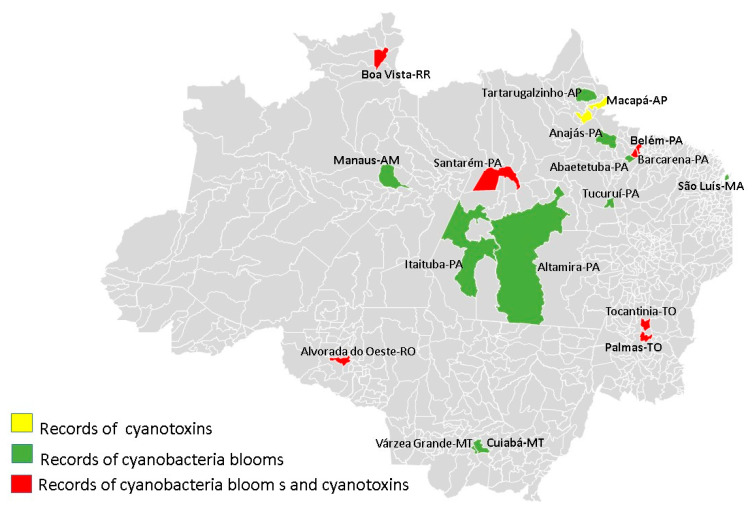
Map showing the records of blooms and cyanotoxins in the Brazilian Legal Amazon.

**Figure 4 toxins-17-00296-f004:**
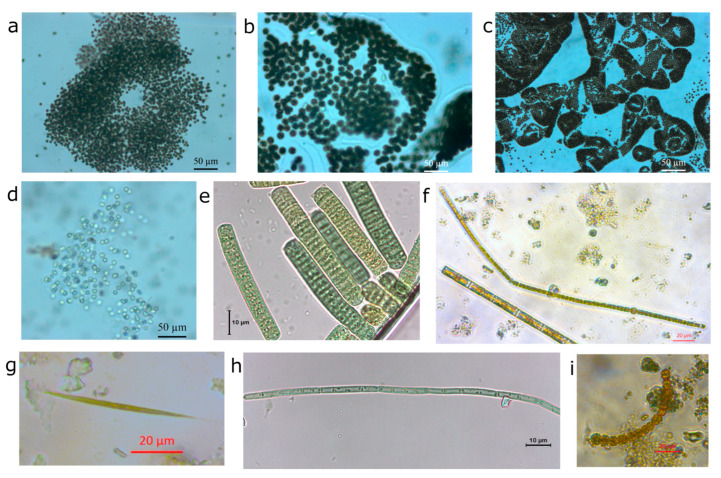
Main species of cyanobacteria present in the blooms. (**a**) *Microcystis aeruginosa* (Kützing) Kützing (**b**) and (**c**) *Microcystis wesenbergii* (Komárek) Komárek *ex* Komárek; (**d**) *Microcystis protocystis* W.B.Crow, colony with *Pseudanabaena mucicola* (Naumann & Huber-Pestalozzi) Schwabe in the mucilage; (**e**) *Dolichospermum* sp.; (**f**) *Anabaena* sp.; (**g**) *Limnothrix planctonica* (Woloszynska) Meffert; (**h**) *Raphidiopsis* sp.; and (**i**) *Phormidium* sp.

**Figure 5 toxins-17-00296-f005:**
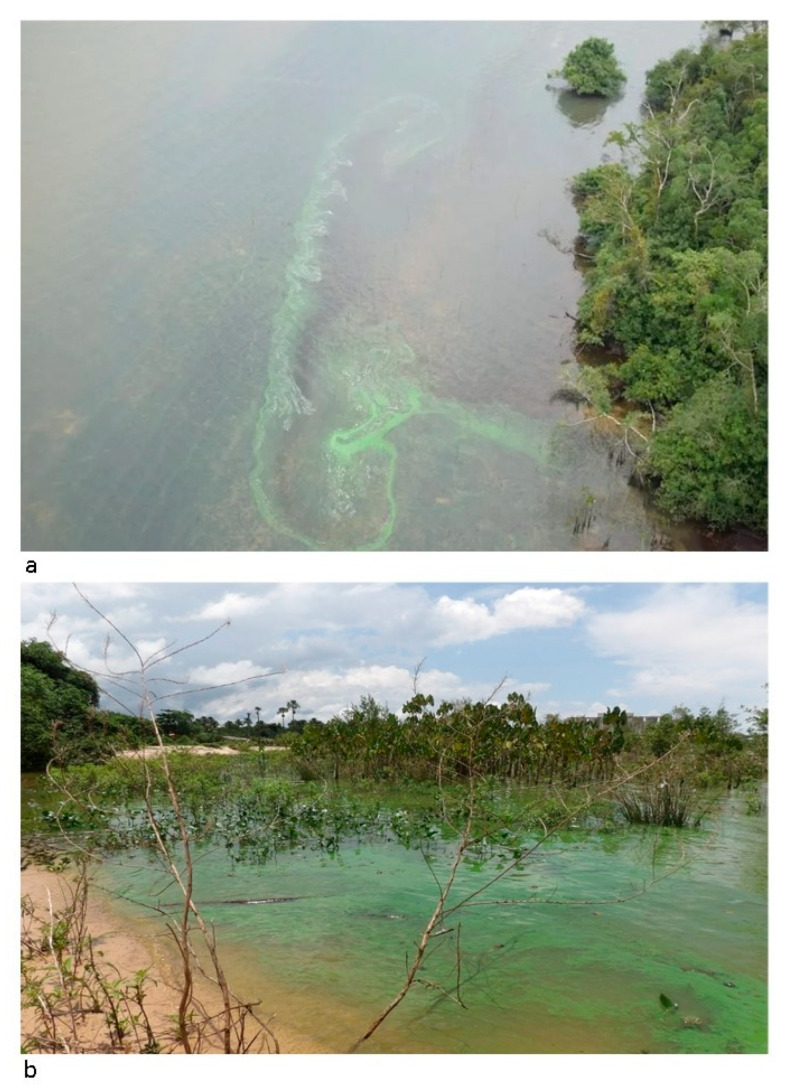
Blooms that occurred in the Pará River, from January to March 2021: (**a**) aerial view of the bloom event, photo from the Evandro Chagas Institute, 2021 [60]; (**b**) frontal view of the bloom event, photo from the Evandro Chagas Institute, 2021 [46].

**Figure 6 toxins-17-00296-f006:**
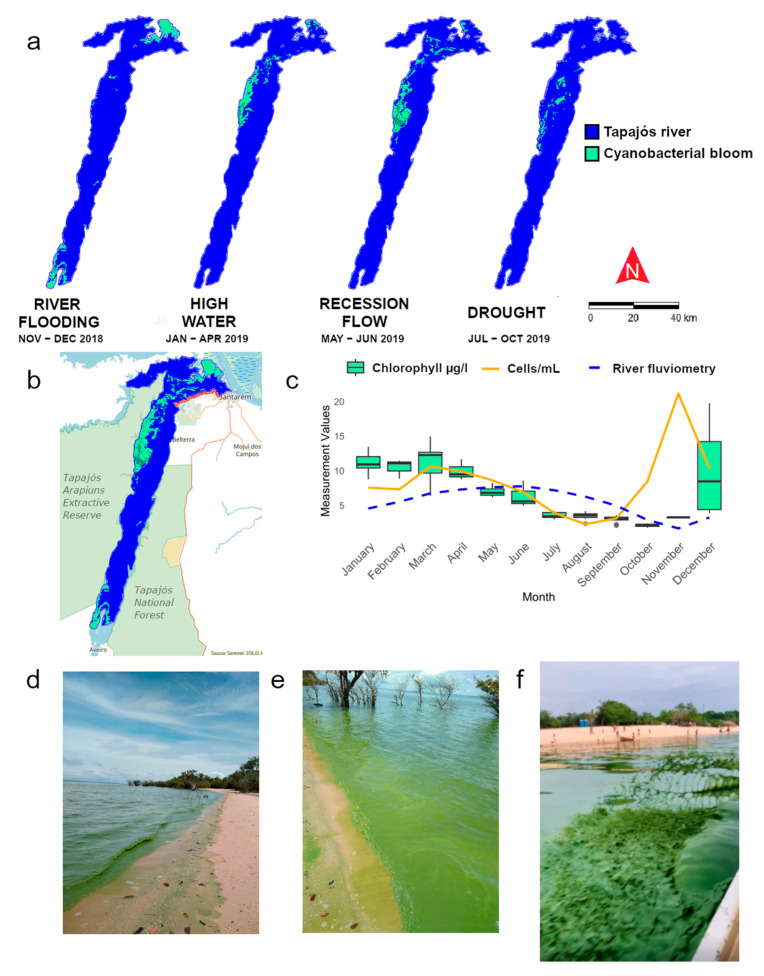
Bloom events in the Tapajós River: (**a**) spatial pattern of the bloom (green spot) over the months and corresponding Amazonian seasonal periods based on remote sensing, with data covering the period from November 2018 to October 2019; (**b**) delineation of the total bloom area from the overlap of the green spots during that time; (**c**) relationship between chlorophyll (green boxes), cells per mL (yellow line), and river fluviometry (blue dashed line) over the months (metrics obtained through CyanoLakes); (**d**,**e**) bloom event that occurred in February 2020, observed in Ponta do Carapanari, urban area of Santarém, Pará; (**f**) bloom event that occurred in January 2025, also observed in Alter do Chão, beach area and district Santarém, Pará.

**Table 1 toxins-17-00296-t001:** The most frequent cyanobacterial genera associated with cyanotoxins and blooms in the analyzed locations.

*Cyanobacteria Genera*	Acre	Amapá	Amazonas	Maranhão	Mato Grosso	Pará	Rondônia	Roraima	Tocantins
*Anabaena*		X				X			
*Aphanizomenon*		X							
*Dolichospermum*		X	X			X			
*Lyngbya*						X			
*Limnothrix*		X							
*Microcystis*		X	X	X	X	X	X		X
*Nostoc*						X			
*Phormidium*						X			
*Planktolyngbya*		X				X	X		X
*Planktothrix*		X	X		X	X	X	X	
*Radiocystis*						X			X
*Raphidiopsis*		X			X	X	X		X
*Sinechocystis*						X			

**Table 2 toxins-17-00296-t002:** Detailed information on toxins recorded in the Brazilian Legal Amazon: toxin detection methods, concentration, presence/absence of bloom, producing genus and references. CYN: cilindrospermopsin; MC: total microcistin; MC-LR: microcistin-LR; STX: saxitoxin.

Brazil State	Toxin	Maximum Concentration of Toxins	Method	Estimation	Dominant Toxin-Producing Genera	Reference
Pará	STX	1.5 µg∙mL^−1^	HPLC	Visual detection of bloom	*Microcystis* sp. *Raphidiopsis* sp.	[47]
Pará	MC	1.25 µg∙mL^−1^	ELISA	20,000 cells∙mL^−1^ Visual detection of bloom	*Microcystis viridis* *Radiocystis fernandoii*	[23,38]
Tocantins	MC, STX, CYN	0.2 µg∙mL^−1^; 0.07 µg∙mL^−1^; 1.1 µg∙mL^−1^	ELISA	2486 cells.mL^−1^	*Raphidiopsis raciborskii* *Planktolyngbya limnetica*	[59]
Pará	MC-LR	12.39 µg∙mL^−1^	HPLC	Visual detection of bloom	*Dolichospermum* sp.	[24]
Roraima	-	-	Toxicological test on mice	164.22 µg∙mL^−1^ chlo-a Visual detection of bloom	*Planktothrix agardhii*	[54]
Rondônia *	MC	0.75 µg∙mL^−1^ (1.26 µg∙mL^−1^)	ELISA	3.2 µg∙mL^−1^ chlo-a 1030 cells.mL^−1^	*Microcystis panniformes*	[50]
Pará	MC-LR	0.1 µg∙mL^−1^	ELISA	2169 cells.mL^−1^ 2.06 µg∙mL^−1^ chlo-a	*Microcystis* 933 cells.mL^−1^	[29,42]
Amapá	MC	1.73 µg∙mL^−1^	ELISA	146.82 µg∙mL^−1^ chlo-a	*Microcystis aeruginosa*	[13]
Pará	MC	0.17 µg∙mL^−1^	ELISA	-	-	[35]
Amapá *	MC-LR	2.1 µg∙mL^−1^	ELISA	1090 cells.mL^−1^, 0.02 mm^3^∙mL^−1^ biovolume	*Limnothrix planctonica* 898.4 cells.mL^−1^; 0.02 mm^3^∙mL^−1^	[14,15]
Pará	MC-LR	1.35 µg∙mL^−1^	HPLC	37,889 cells.mL^−1^	*Dolichospermum* sp.	[14]
Pará	MC	11.95 µg∙mL^−1^	HPLC	3209 ind.mL^−1^	*Dolichospermum* sp.	[48]

* Cyanotoxins in the absence of cyanobacteria blooms.

## Data Availability

The original contributions presented in this study are included in the article/Appendix A. Further inquiries can be directed to the corresponding author(s).

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
