# Peer review of "Cyanobacterial Blooms and the Presence of Cyanotoxins in the Brazilian Amazon"

_toxins, 2025, doi:10.3390/toxins17060296_

Round 1
Reviewer 1 Report
Comments and Suggestions for Authors
In this paper, the authors conduct a systematic literature review to assess the geographic distribution, frequency, and toxin-producing potential of cyanobacterial blooms in the Brazilian Amazon. They examine 52 studies and identify 155 cyanobacterial taxa across nine states, with Microcystis and Planktothrix being the most frequently reported genera. The review reveals a growing incidence of blooms and toxin detection in several states while also highlighting major data gaps. The authors argue that increased monitoring, improved analytical infrastructure, and consideration of climate change impacts are urgently needed for better public health and ecosystem management.
While this paper provides a valuable synthesis of regional data on cyanobacteria and their toxins, it lacks some key limitations:
1) The criteria for defining blooms are inconsistently applied across studies, which complicates interpretation and undermines comparisons.
2) Although the authors emphasize the urgency of improved monitoring, they fall short of offering actionable recommendations or prioritization strategies.
3) The Discussion reiterates known issues (e.g., low monitoring capacity, lack of data) but misses opportunities to critically assess methodological biases in the literature, such as an over-reliance on visual bloom detection or the exclusion of grey literature.
4) The literature search strategy does not appear to be fully transparent or reproducible; specifically I am referring to inclusion/exclusion criteria and search terms for each state.
Specific line feedback (L#) is provided below:
L67–71: Consider explicitly stating research questions or review goals beyond “expand the existing database.” What are the spatial patterns in toxin-producing species? Where are the critical data gaps?
L244–274: Your call for more monitoring is valid but reads as generic. I suggest adding specific recommendations (e.g., prioritize remote sensing in unmonitored floodplains or establish early-warning thresholds based on chlorophyll-a or toxin levels). Consult relevant literature and cite them when making these suggestions.
L156–160: The manuscript acknowledges inconsistencies in bloom definitions but does not resolve them or offer harmonized criteria. I recommend clarifying how you handled discrepancies when comparing across studies.
L276–302: The search strategy is under-detailed. You should list exact keywords used per state, clarify how grey literature was assessed for quality, and specify whether non-peer-reviewed materials were weighted equally.
Given that Microcystis was the most frequently detected cyanobacterial genus in the reviewed studies, you should consider expanding the Discussion to situate this finding within the broader global context. Microcystis is a globally dominant bloom-forming cyanobacterium known for its production of microcystins, which pose significant health risks to both humans and animals. Several reviews (some suggested below as a reference) provide insights into its ecological adaptability, global distribution, and toxicological impacts. Including this perspective would enhance the international relevance of the study and underscore the urgency of improved monitoring and management in the Amazon. You could suggest placing this around L192–212 in the Discussion, where the manuscript discusses Microcystis dominance in Pará and Lago Grande do Curuai, or as part of the Conclusions (L253–275) to frame the broader implications.
-Harke, M. J., Steffen, M. M., Gobler, C. J., Otten, T. G., Wilhelm, S. W., Wood, S. A., & Paerl, H. W. (2016). A review of the global ecology, genomics, and biogeography of the toxic cyanobacterium, Microcystis spp. Harmful algae, 54, 4-20. https://doi.org/10.1016/j.hal.2015.12.007
-Dick, G. J., Duhaime, M. B., Evans, J. T., Errera, R. M., Godwin, C. M., Kharbush, J. J., ... & Denef, V. J. (2021). The genetic and ecophysiological diversity of Microcystis. Environmental Microbiology, 23(12), 7278-7313. https://doi.org/10.1111/1462-2920.15615
-Shahmohamadloo, R. S., Frenken, T., Rudman, S. M., Ibelings, B. W., & Trainer, V. L. (2023). Diseases and disorders in fish due to harmful algal blooms. In Climate Change on Diseases and Disorders of Finfish in Cage Culture (pp. 387-429). GB: CABI. https://doi.org/10.1079/9781800621640.0010
Author Response
Comment 1: The criteria for defining blooms are inconsistently applied across studies, which complicates interpretation and undermines comparisons.
Response 1: We agree that the comparisons between studies were complicated, so we decided to adopt the quantitative bloom definition proposed by Frau [35] in an extensive review paper to propose a quantitative definition of cyanobacteria bloom for different types of water (oligotrophic to eutrophic). We believe that the threshold proposed for oligotrophic waters fits the Amazonian Water bodies very well. These definitions are more precise than the WHO alert levels (adopted in the previous version), which do not specify the exact threshold for considering a bloom. In the article: “As there is no scientific consensus to define a cyanobacterial bloom, we adopted the quantitative bloom definition proposed by Frau [35] in a revision of papers with cyanobacterial blooms worldwide. In this context, we considered Amazonian water bodies as oligotrophic inland waters, and the thresholds for considering a cyanobacteria bloom in this study were: cyanobacteria cell density >1,615 cells mL-1, chlorophyll-a >0.32 µg L-1, biovolume >0.57mm³ L-1 [35]. We also considered the papers containing qualitative detection, such as visual identification of greenish spots and/or the presence of foam [35]. A recent review of papers on cyanobacteria blooms conducted in Latin America showed definitions of bloom varying from 2,000 to 50,000 cells mL-1, with 2,000 cells mL-1 being the most frequent definition of bloom, as reported in 47% of the papers [29]. Both results present similar quantitative numbers to define cyanobacteria blooms. The spatiotemporal monitoring of the blooms for the Tapajós River was carried out using the CyanoLakes platform (www.cyanolakes.com).”
Comment 2: Although the authors emphasize the urgency of improved monitoring, they fall short of offering actionable recommendations or prioritization strategies.
Response 2: After revision, we proposed a monitoring strategy that does not require specialists and is also recommended in the Ministry of Health guidance [36]: to proceed with weekly monitoring of cyanotoxins in raw water (WTP inlet) as an alternative to monitoring cyanobacteria and chlorophyll-a. We recommend that monitoring in the drinking water catchment points be done by analyzing cyanotoxins instead of cyanobacteria identification and counting or chlorophyll a. Cyanotoxin analyses can be done by ELISA, which requires buying an ELISA reader and washer and then ELISA kits. These are simple and rapid methods that allow the processing of a large number of samples in a short time. Aguilera et al. [29] also recommend prioritizing toxin analyses by the ELISA method. In the article: “The lack of cyanotoxin data in the Amazon (cited by SISAGUA) is a growing concern given the current trend of worsening water quality worldwide. For this reason, we recommend that, as a strategy for the Amazon region, monitoring at drinking water intake points be done through cyanotoxin analysis rather than through the identification and counting of cyanobacteria or chlorophyll a. Cyanotoxin analyses can be performed by ELISA, which requires the purchase of an ELISA reader and washer, and then ELISA kits. These are simple and rapid methods that allow the processing of a large number of samples in a short time. Aguillera et al. 2023 also recommend prioritizing toxin analyses by the ELISA method. The Ministry of Health's ordinance (reference xx) also allows for the direct analysis of toxins, eliminating the need to identify and count cyanobacteria and chlorophyll. We believe that this is the best strategy for the Amazon region, corroborated by Aguillera [xx]. This recommendation will overcome the need for specialized technicians to identify genera and count cyanobacteria cells”.
Comment 3: The Discussion reiterates known issues (e.g., low monitoring capacity, lack of data) but misses opportunities to critically assess methodological biases in the literature, such as an overreliance on visual bloom detection or the exclusion of grey literature.
Response 3: All the grey literature, such as journalistic articles and podcasts, was excluded. The criteria for literature inclusion were published articles, theses, dissertations, books, and technical reports with information on cyanobacteria blooms, toxins, or both in water bodies in the Brazilian Amazon.” Regarding the methodological biases in the literature, after revision, we decided to define thresholds for cyanobacterial blooms according to recently published papers. This article is an excellent opportunity to discuss the Amazonian data on cyanobacteria and classify them as either blooms or not. We also proposed an alternative form of monitoring that does not require specialists, as recommended in the Ministry of Health guidance [36]: to proceed with weekly monitoring of cyanotoxins in raw water (WTP inlet) as an alternative to monitoring cyanobacteria and chlorophyll-a.
Comment 4: The literature search strategy does not appear to be fully transparent or reproducible; specifically, I am referring to inclusion/exclusion criteria and search terms for each state.
Response 4: We agree with the reviewer and have completely restructured the Materials and Methods section to make it more reproducible. We added links to the repositories of scientific institutions in the Brazilian Amazon, which contain most of the regional studies. We also set the inclusion/exclusion criteria. In the article: “For this work, a bibliographic review was conducted, adhering to the criteria of a systematic review, which involves a predefined set of criteria concerning the type of study [50]. We searched various databases, including Google Scholar [51], Scielo [52], and Periódicos Capes [53], as well as repositories of scientific institutions in the Brazilian Amazon [54-62]. The search combinations were: “Cyanophyta and Bloom and Amazon”; “Cyanobacteria and Bloom”, “Phytoplankton”, “Floração + cianobactérias”, "toxic cyanobacteria," "cyanotoxins," "cianotoxinas," "cianobactéria tóxica," and "cyanoHAB”. Since scientific data on cyanobacteria and cyanotoxins in the Amazon Region are scarce and dispersed, we also read papers to look for, in their references, more scientific studies on cyanobacteria blooms and cyanotoxins in the Region. The selection process involved three stages: analyzing titles, reading abstracts, and conducting complete readings. The criteria for literature inclusion were published articles, theses, dissertations, books, conference presentations, and technical reports with information on cyanobacteria blooms, toxins, or both in water bodies in the Brazilian Amazon. Each article was analyzed, with those that did not report cyanobacteria blooms or toxins in their results being discarded.”
Comment 5: Specific line feedback (L#): L67–71: Consider explicitly stating research questions or review goals beyond “expand the existing database.” What are the spatial patterns in toxin-producing species? Where are the critical data gaps?
Response 5: We reviewed our goals. In the paper: “In this context, this study aimed to verify, through a literature review, the distribution of cyanobacterial blooms and toxins in the states that comprise the Brazilian Legal Amazon, and to identify spatial-temporal patterns associated with the rise in cyanobacteria density. It also provided recommendations for cyanotoxin monitoring in the Brazilian Amazon.”
Comment 6: Specific line feedback (L#): L244–274: Your call for more monitoring is valid but reads as generic. I suggest adding specific recommendations (e.g., prioritize remote sensing in unmonitored floodplains or establish early-warning thresholds based on chlorophyll-a or toxin levels). Consult relevant literature and cite them when making these suggestions.
Response 6: After revision, we proposed a monitoring strategy that does not require specialists and is also recommended in the Ministry of Health guidance (GM/MS 888/2021): to proceed with weekly monitoring of cyanotoxins in raw water (WTP inlet) as an alternative to monitoring cyanobacteria and chlorophyll-a. We recommend that monitoring in the drinking water catchment points should be done by analyzing cyanotoxins instead of cyanobacteria identification and counting or chlorophyll a. Cyanotoxin analyses can be done by ELISA, which requires buying an ELISA reader and washer and then ELISA kits. These are simple and rapid methods that allow the processing of a large number of samples in a short time. Aguilera et al. 2023 also recommend prioritizing toxin analyses in Latin America by the ELISA method. In the paper: “The Brazilian Amazon boasts rich biodiversity, yet scientific research and monitoring resources remain scarce. Numerous articles estimate the region's phytoplankton biomass using remote sensing to gauge chlorophyll-a, but few also incorporate data on biovolume or cyanobacteria counts. For this reason, we recommend that, as a strategy for the Amazon region, monitoring at drinking water intake points be done through cyanotoxin analysis rather than through the identification and counting of cyanobacteria or chlorophyll a. Cyanotoxin analyses can be performed by ELISA, which requires the purchase of an ELISA reader and washer, and then ELISA kits. These are simple and rapid methods that allow the processing of a large number of samples in a short time. Aguillera et al. [29] also recommend prioritizing toxin analyses by the ELISA method. The Ministry of Health's ordinance [36] also permits the direct analysis of toxins, thereby eliminating the need to identify and count cyanobacteria and chlorophyll. We believe that this is the best strategy for the Amazon region, corroborated by [29]. This recommendation will overcome the need for specialized technicians to identify genera and count cyanobacteria cells.”
Comment 7: Specific line feedback (L#): L156–160: The manuscript acknowledges inconsistencies in bloom definitions but does not resolve them or offer harmonized criteria. I recommend clarifying how you handled discrepancies when comparing across studies.
Response 7: We agree that the comparisons between studies were complicated, so we decided to adopt the quantitative bloom definition proposed by Frau [35] in an extensive review paper to propose a quantitative definition of cyanobacteria bloom for different types of water (oligotrophic to eutrophic). We believe the threshold proposed for oligotrophic waters fits the Amazonian Water bodies very well. These definitions are more precise than the WHO alert levels (adopted in the previous version), which do not specify the exact threshold for considering a bloom. In the paper: “As there is no scientific consensus on defining a cyanobacterial bloom, we adopted the quantitative bloom definition proposed by Frau [35] in revising papers on cyanobacterial blooms worldwide. In this context, we considered Amazonian water bodies as oligotrophic inland waters, and the thresholds for considering a cyanobacteria bloom in this study were: cyanobacteria cell density >1,615 cells mL-1, chlorophyll-a >0.32 µg L-1, biovolume >0.57mm³ L-1 [35]. We also considered the papers containing qualitative detection, such as visual identification of greenish spots and/or the presence of foam. A recent review of papers on cyanobacteria blooms conducted in Latin America showed definitions of bloom varying from 2,000 to 50,000 cells mL-1, with 2,000 cells mL-1 being the most frequent definition of bloom, as reported in 47% of the papers [29]. Both results present similar quantitative numbers to define cyanobacteria blooms. The spatiotemporal monitoring of the blooms for the Tapajós River was carried out using the CyanoLakes platform.”
Comment 8: Specific line feedback (L#): L276–302: The search strategy is underdetailed. You should list exact keywords used per state, clarify how grey literature was assessed for quality, and specify whether non-peer-reviewed materials were weighted equally.
Response 8: We agree with the reviewer and have wholly restructured the Material and Methods section to make it more reproducible. We added links to the repositories of scientific institutions in the Brazilian Amazon, which contain most regional studies. We also set the inclusion/exclusion criteria. Moreover, all the grey literature, such as journalistic articles and podcasts, was excluded. The criteria for literature inclusion were published articles, theses, dissertations, books, and technical reports with information on cyanobacteria blooms, toxins, or both in water bodies in the Brazilian Amazon. In the paper: “For this work, a bibliographic review was conducted, adhering to the criteria of a systematic review, which involves a predefined set of criteria concerning the type of study [50]. We searched various databases, including Google Scholar [51], Scielo [52], and Periódicos Capes [53], as well as repositories of scientific institutions in the Brazilian Amazon [54-62]. The search combinations were: “Cyanophyta and Bloom and Amazon”; “Cyanobacteria and Bloom”, “Phytoplankton”, “Floração + cianobactérias”, "toxic cyanobacteria," "cyanotoxins," "cianotoxinas," "cianobactéria tóxica," and "cyanoHAB”. Since scientific data on cyanobacteria and cyanotoxins in the Amazon Region are scarce and dispersed, we also read papers to look for, in their references, more scientific studies on cyanobacteria blooms and cyanotoxins in the Region. The selection process involved three stages: analyzing titles, reading abstracts, and conducting complete readings. The criteria for literature inclusion were published articles, theses, dissertations, books, conference presentations, and technical reports with information on cyanobacteria blooms, toxins, or both in water bodies in the Brazilian Amazon. Each article was analyzed, with those that did not report cyanobacteria blooms or toxins in their results being discarded.”
Comment 9: Given that Microcystis was the most frequently detected cyanobacterial genus in the reviewed studies, you should consider expanding the Discussion to situate this finding within the broader global context. Microcystis is a globally dominant bloom-forming cyanobacterium known for its production of microcystins, which pose significant health risks to both humans and animals. Several reviews (some suggested below as a reference) provide insights into its ecological adaptability, global distribution, and toxicological impacts. Including this perspective would enhance the international relevance of the study and underscore the urgency of improved monitoring and management in the Amazon. You could suggest placing this around L192–212 in the Discussion, where the manuscript discusses Microcystis dominance in Pará and Lago Grande do Curuai, or as part of the Conclusions (L253–275) to frame the broader implications.
-Harke, M. J., Steffen, M. M., Gobler, C. J., Otten, T. G., Wilhelm, S. W., Wood, S. A., & Paerl, H. W. (2016). A review of the global ecology, genomics, and biogeography of the toxic cyanobacterium, Microcystis spp. Harmful algae, 54, 4-20. https://doi.org/10.1016/j.hal.2015.12.007
-Dick, G. J., Duhaime, M. B., Evans, J. T., Errera, R. M., Godwin, C. M., Kharbush, J. J., ... & Denef, V. J. (2021). The genetic and ecophysiological diversity of Microcystis. Environmental Microbiology, 23(12), 7278-7313. https://doi.org/10.1111/1462-2920.15615
-Shahmohamadloo, R. S., Frenken, T., Rudman, S. M., Ibelings, B. W., & Trainer, V. L. (2023). Diseases and disorders in fish due to harmful algal blooms. In Climate Change on Diseases and Disorders of Finfish in Cage Culture (pp. 387-429). GB: CABI. https://doi.org/10.1079/9781800621640.0010
Response 9: “The most common bloom-forming genera for the Brazilian Legal Amazon were Microcystis and Planktothrix; they are also very common in Latin American countries [29]. Both produce microcystin, the most common cyanotoxin found in the papers (Table 2).”
Reviewer 2 Report
Comments and Suggestions for Authors
The study addresses a critical gap in cyanobacterial bloom research in the understudied Brazilian Amazon and provides a comprehensive review of historical and recent bloom events, toxin detection, and regional variability.
I have made the following shortcomings in the manuscript which need to be addessed.
The ‘systematic review’ claim needs stronger justification. The methodology description is brief and lacks details on search strategy, inclusion/exclusion criteria, and quality assessment of included studies.
The inclusion of journalistic articles and podcasts alongside peer-reviewed research raises concerns about the rigor of the review. The authors must indicate the level of quality control of each kind of document.
The adoption of WHO alert levels for bloom definition needs more critical discussion in the Amazonian context. Local calibration is mentioned but not thoroughly addressed.
The discussion often mixes descriptive accounts with analytical interpretations, making it difficult to discern robust conclusions.
The use of the term ‘bloom’ is highly problematic. The authors admit that in many cases, the definition of a bloom is not met. This must be addressed throughout the document.
The discussion on the drivers of bloom formation (e.g., deforestation, mining, climate change) is superficial. More in-depth analysis of these factors is needed.
The public health implications of cyanotoxins are mentioned but not explored in detail.
The paper lacks a strong statistical analysis of the data.
The English is generally understandable but contains numerous grammatical errors and difficult phrasing. Careful editing is required.
There are many instances of missing articles (the, a, an) and incorrect word usage.
The manuscript has the potential to contribute valuable information to the field. However, significant revisions are needed to address the methodological weaknesses, improve the data analysis and presentation, deepen the discussion, and refine the English language.
Comments on the Quality of English LanguageThe English is generally understandable but contains numerous grammatical errors and difficult phrasing. Careful editing is required.
There are many instances of missing articles (the, a, an) and incorrect word usage.
Author Response
Comment 1: The ‘systematic review’ claim needs stronger justification. The methodology description is brief and lacks details on the search strategy, inclusion/exclusion criteria, and quality assessment of included studies.
Response 1: We agree with the reviewer, and we have completely restructured the Materials and Methods section to make it more reproducible and to provide more details on the search strategy. We added links to the repositories of scientific institutions in the Brazilian Amazon, which contain most of the regional studies. We also set the inclusion/exclusion criteria. In the paper: “For this work, a bibliographic review was conducted, adhering to the criteria of a systematic review, which involves a predefined set of criteria concerning the type of study [50]. We searched various databases, including Google Scholar [51], Scielo [52], and Periódicos Capes [53], as well as repositories of scientific institutions in the Brazilian Amazon [54-62]. The search combinations were: “Cyanophyta and Bloom and Amazon”; “Cyano-bacteria and Bloom”, “Phytoplankton”, “Floração + cianobactérias”, "toxic cyanobacteria," "cyanotoxins," "cianotoxinas," "cianobactéria tóxica," and "cyanoHAB”. Since scientific data on cyanobacteria and cyanotoxins in the Amazon Region are scarce and dispersed, we also read papers to look for, in their references, more scientific studies on cyanobacteria blooms and cyanotoxins in the Region. The selection process involved three stages: analyzing titles, reading abstracts, and conducting complete readings. The criteria for literature inclusion were published articles, theses, dissertations, books, conference presentations, and technical reports with information on cyanobacteria blooms, toxins, or both in water bodies in the Brazilian Amazon. Each article was analyzed, with those that did not report cyanobacteria blooms or toxins in their results being discarded.”
Comment 2: The inclusion of journalistic articles and podcasts alongside peer-reviewed research raises concerns about the rigor of the review. The authors must indicate the level of quality control of each kind of document.
Response 2: All the grey literature, such as journalistic articles and podcasts, was excluded. The criteria for literature inclusion were published articles, theses, dissertations, books, and technical reports with information on cyanobacteria blooms, toxins, or both in water bodies in the Brazilian Amazon.”
Comment 3: The adoption of WHO alert levels for bloom definition needs more critical discussion in the Amazonian context. Local calibration is mentioned but not thoroughly addressed.
Response 3: We agree that the WHO alert levels (adopted in the previous version) need more critical discussion, and they do not specify the exact threshold for considering a bloom. For this reason, we decided to adopt the quantitative bloom definition proposed by Frau [35] in a comprehensive review paper, with the aim of proposing a quantitative definition of cyanobacteria blooms for different types of water, ranging from oligotrophic to eutrophic. We believe that the threshold proposed for oligotrophic waters fits very well with the Amazonian Water bodies. In the paper: “As there is no scientific consensus on defining a cyanobacterial bloom, we adopted the quantitative bloom definition proposed by Frau [35] in revising papers on cyanobacterial blooms worldwide. In this context, we considered Amazonian water bodies as oligo-trophic inland waters, and the thresholds for considering a cyanobacteria bloom in this study were: cyanobacteria cell density >1,615 cells mL-1, chlorophyll-a >0.32 µg L-1, biovolume >0.57mm³ L-1 [35]. We also considered the papers containing qualitative detection, such as visual identification of greenish spots and/or the presence of foam. A recent review of papers on cyanobacteria blooms conducted in Latin America showed definitions of bloom varying from 2,000 to 50,000 cells mL-1, with 2,000 cells mL-1 being the most frequent definition of bloom, as reported in 47% of the papers [29]. Both results present similar quantitative numbers to define cyanobacteria blooms. The spatiotemporal monitoring of the blooms for the Tapajós River was carried out using the CyanoLakes platform.
Comment 4: The discussion often mixes descriptive accounts with analytical interpretations, making it difficult to discern robust conclusions.
Response 4: After we defined the bloom threshold this problem was solved.
Comment 5: The use of the term ‘bloom’ is highly problematic. The authors admit that in many cases, the definition of a bloom is not met. This must be addressed throughout the document.
Response 5: We agreed, and after the revision, we addressed throughout the document the same bloom definition, according to Frau 2023 (described in the Materials and Methods section).
Comment 6: The discussion on the drivers of bloom formation (e.g., deforestation, mining, climate change) is superficial. More in-depth analysis of these factors is needed.
Response 6: We decided to focus more on the search for patterns for cyanobacteria blooms. The blooms' drivers need more profound research, which is not the aim of the present paper.
Comment 7: The public health implications of cyanotoxins are mentioned but not explored in detail.
Response 7: We removed the mention of public health implications and changed it to: “This information may be used by drinking water treatment plants, reservoirs, and others to diminish health impacts.”
Comment 8: The paper lacks a strong statistical analysis of the data.
Response 8: As a review paper, we did not use much numerical data. This kind of information may be used with strong statistical analysis for future review papers with data for the Amazonian Floodplain Lakes, which contains a lot of cyanobacteria, but it is material for other specific studies.
Comment 9: The English is generally understandable but contains numerous grammatical errors and difficult phrasing. Careful editing is required.
Response 9: We agreed and hired a qualified translator to review the article. The translator checked the text and made improvements in the English language. We hope that this version's English is satisfactory and on the level of publication.
Comment 10: The manuscript has the potential to contribute valuable information to the field. However, significant revisions are needed to address the methodological weaknesses, improve the data analysis and presentation, deepen the discussion, and refine the English language.
Response 10: We agreed with the reviewer, and we dramatically improved the paper by refining the English language and changing the whole methodological section.
Round 2
Reviewer 1 Report
Comments and Suggestions for Authors
The manuscript is ready for publication.
Author Response
Comment: The manuscript is ready for publication.
Response: Thank you!
Reviewer 2 Report
Comments and Suggestions for Authors
Only minor grammatical errors persist, which can be rectified during the final proofreading stage, contingent upon acceptance.
Author Response
Comment: Only minor grammatical errors persist, which can be rectified during the final proofreading stage, contingent upon acceptance.
Response: We have reviewed this version and made some grammatical corrections to refine the text (Page 1, line 8; Page 2, line 58; Page 9, lines 138-141; Page 13, line 230; Page 13, line 233; Page 16, Table 2.
Thank you for your valuable suggestions.